# ERK Signaling Pathway Is Constitutively Active in NT2D1 Non-Seminoma Cells and Its Inhibition Impairs Basal and HGF-Activated Cell Proliferation

**DOI:** 10.3390/biomedicines11071894

**Published:** 2023-07-04

**Authors:** Luisa Gesualdi, Marika Berardini, Bianca Maria Scicchitano, Clotilde Castaldo, Mariano Bizzarri, Antonio Filippini, Anna Riccioli, Chiara Schiraldi, Francesca Ferranti, Domenico Liguoro, Rita Mancini, Giulia Ricci, Angela Catizone

**Affiliations:** 1Section of Histology and Medical Embryology, Department of Anatomy, Histology, Forensic-Medicine and Orthopedics, “Sapienza” University of Rome, 00161 Rome, Italy; luisa.gesualdi@uniroma1.it (L.G.); marika.berardini@uniroma1.it (M.B.); antonio.filippini@uniroma1.it (A.F.); anna.riccioli@uniroma1.it (A.R.); 2Section of Histology, Department of Life Sciences and Public Health, Università Cattolica del Sacro Cuore, 00168 Rome, Italy; biancamaria.scicchitano@unicatt.it; 3Fondazione Policlinico Universitario A. Gemelli IRCCS, 00168 Rome, Italy; 4Department of Public Health, University of Naples Federico II, 80131 Naples, Italy; clotilde.castaldo@unina.it; 5Department of Experimental Medicine, “Sapienza” University of Rome, 00161 Rome, Italy; mariano.bizzarri@uniroma1.it; 6Systems Biology Group Lab, 00161 Rome, Italy; 7Department of Experimental Medicine, Università degli Studi della Campania “Luigi Vanvitelli”, 80138 Naples, Italy; chiara.schiraldi@unicampania.it; 8Human Spaceflight and Scientific Research Unit, Italian Space Agency, 00133 Roma, Italy; francesca.ferranti@asi.it; 9Department of Clinical and Molecular Medicine, Sant’Andrea Hospital, “Sapienza” University of Rome, 00185 Rome, Italy; domenico.liguoro@uniroma1.it (D.L.); rita.mancini@uniroma1.it (R.M.)

**Keywords:** c-Met/HGF system, testicular germ cell tumors, MAPK/ERK pathway, tumor microenvironment

## Abstract

c-MET/hepatocyte growth factor (HGF) system deregulation is a well-known feature of malignancy in several solid tumors, and for this reason this system and its pathway have been considered as potential targets for therapeutic purposes. In previous manuscripts we reported c-MET/HGF expression and the role in testicular germ cell tumors (TGCTs) derived cell lines. We demonstrated the key role of c-Src and phosphatidylinositol 3-kinase (PI3K)/AKT adaptors in the HGF-dependent malignant behavior of the embryonal carcinoma cell line NT2D1, finding that the inhibition of these onco-adaptor proteins abrogates HGF triggered responses such as proliferation, migration, and invasion. Expanding on these previous studies, herein we investigated the role of mitogen-activated protein kinase (MAPK)/extracellular signal regulated kinase (ERK) pathways in the HGF-dependent and HGF-independent NT2D1 cells biological responses. To inhibit MAPK/ERK pathways we chose a pharmacological approach, by using U0126 inhibitor, and we analyzed cell proliferation, collective migration, and chemotaxis. The administration of U0126 together with HGF reverts the HGF-dependent activation of cell proliferation but, surprisingly, does not exert the same effect on NT2D1 cell migration. In addition, we found that the use of U0126 alone significantly promotes the acquisition of NT2D1 «migrating phenotype», while collective migration of NT2D1 cells was stimulated. Notably, the inhibition of ERK activation in the absence of HGF stimulation resulted in the activation of the AKT-mediated pathway, and this let us speculate that the paradoxical effects obtained by using U0126, which are the increase of collective migration and the acquisition of partial epithelium–mesenchyme transition (pEMT), are the result of compensatory pathways activation. These data highlight how the specific response to pathway inhibitors, should be investigated in depth before setting up therapy.

## 1. Introduction

It is well known that tyrosine-kinase receptors and associated signaling pathways are often involved in the onset and progression of many types of tumors. Among them, the c-MET pathway, activated by the hepatocyte growth factor (HGF), is one of the most studied [1]. Moreover, because of its association with cancer progression, it has been considered a potential target for therapeutic purposes in several solid cancers [2,3].

In previous papers we analyzed c-MET and HGF expression in biopsies derived from patients affected by testicular germ cell tumors (TGCTs), concluding that non-seminomas (embryonal carcinoma, yolk sac tumor, and teratoma) present the highest expression level of this receptor on the plasma membrane [4] and, besides this, embryonal carcinoma has the highest expression of HGF in the tumor microenvironment with respect to seminoma lesions [5]. We studied c-MET and HGF expression even in seminoma and non-seminoma derived cell lines (TCam-2, NCCIT, NT2D1), demonstrating that HGF is not expressed by all these cell lines [4]. Accordingly, previous data from Selfe [6,7] and coworkers demonstrated that the c-MET receptor is expressed but not constitutively activated in the same cell lines. We also demonstrated that the non-seminoma cell line NT2D1 is the best responsive to the administration of exogenous HGF, with respect to TCam-2 and NCCIT cell lines [4]. Notably, NT2D1 cells derive from an embryonal carcinoma lesion which is the lesion from which all non-seminomas take origin. In this cell line, we already demonstrated that c-Src and phosphatidylinositol 3-kinase (PI3K) pharmacological inhibition abrogates the HGF-dependent increase of cell proliferation, polarized and collective migration, as well as cell invasion. We also found that, surprisingly, c-Src or PI3K pharmacological inhibition, when administered in basal culture conditions, increases NT2D1 invasiveness via an HGF-independent way, highlighting the importance of the microenvironmental cues in modulating cellular responses to pharmacological stimuli [5,8].

The aim of this work was to investigate mitogen-activated protein kinase (MAPK)/extracellular signal regulated kinase (ERK) signaling pathway and its involvement in c-MET/HGF signal transduction since its role in the progression of several solid oncological lesions is well known [9]. Indeed, ERK has key roles in the control of cell proliferation, cell cycle regulation and survival, as well as tumor extracellular matrix degradation, and tumor angiogenesis [10,11]. This evidence indicates this pathway to be considered a therapeutic target in many cancer treatments. Raf, MEK1/2, and ERK1/2 are considered important targets to design small molecular weight inhibitors, in particular, MEK1/2 have been relatively advantageous for the design of highly selective ATP-non-competitive inhibitors. In fact, U0126 is a potent ATP- and ERK1/2 non-competitive inhibitor used for scientific research because of its high specificity [12].

It is known in the literature that the PI3K/AKT/mTOR and RAS/RAF/MEK/ERK pathways can interact at different points and trigger cross-activation and cross-inhibition phenomena which in turn may depend on the relative strengths of different growth factors and/or inputs from other pathways. Indeed, data from preclinical studies suggest that tumors are able to use redundant signaling pathways in order to develop resistance as an escaping mechanism from anticancer agents directed to upstream RTKs or to other components of the same pathways [13]. In the present study, we investigated the role of ERK in the HGF-independent and HGF-dependent (c-MET activated) malignant behavior of NT2D1 non-seminoma cells, studying the effects of pharmacological ERK inhibition on the already described biological responses to HGF (proliferation, migration, chemotaxis) [4]. The oncogenic transformation is characterized not only by deregulated growth control but also by pronounced morphological changes resulting from alterations in the organization of the actin cytoskeleton and adhesive interactions [14,15,16]. For these reasons in this study, we analyzed the influence of ERK inhibition on cytoskeleton dynamics and cell–cell and cell–matrix interactions. Cytoskeleton modifications can be related even to the epithelial–mesenchymal transition (EMT) which is a physiological process during embryonic development and tissue regeneration but acquires a pathological meaning during oncogenic transformation, being a hallmark of malignancy of epithelial cancers [17]. Of note, it is well known that the system c-Met/HGF exerts a driving role in both physiological and pathological aspects of EMT, and therefore we considered it important to study the role of HGF-triggered ERK activation in NT2D1 cell phenotypic EMT markers [18,19,20].

## 2. Materials and Methods

### 2.1. Cell Culture

NT2D1 embryonal carcinoma cells (American Type Culture Collection (ATCC) were used in this study and cultured as reported in [4,5,8]. Cells, cultured in DMEM (Sigma Aldrich, cat. D6546, St. Louis, MO, USA) complemented with 10% foetal bovine serum (FBS Gibco, cat. 10270, Gland Island, NY, USA), were treated, when indicated, with 40 ng/mL of HGF (human recombinant HGF, R&D Systems, cat. 294-HG, Minneapolis, MN, USA), 5 µM of MAPK-MEK1/2 selective inhibitor U0126 (U0126-Et-OH, Selleckchem, cat. S1102, USA and Canada only) or both molecules. The U0126 toxicity was evaluated by cell viability FACS analysis, using propidium iodide. We tested different concentrations (2.5, 5, 10, 15 µM) as suggested in the literature. To obtain an increase in G1 phase cell percentage, cells were cultured for 15 h without serum (starvation). Cells were incubated at 37 °C in a humidified atmosphere with 5% CO_2_.

### 2.2. Cell Proliferation Assay

To test the role of ERK on HGF-induced NT2D1 cell proliferation we performed proliferation assays using the MAPK-MEK1/2 selective inhibitor U0126. In this regard, we cultured 9 × 10^4^ NT2D1 cells in 12-well plates (15,000 cell/cm^2^) (Falcon, Lincon Park, NJ, USA) with DMEM 10% FBS. After complete adhesion (24 h), cells were starved under serum-free conditions and after 15 h treated with DMEM 2% FBS (control condition), adding, when indicated, HGF (40 ng/mL, R&D System, Minneapolis, MN, USA) and U0126 5 µM (Selleckchem, Houston, TX, USA), used alone or co-administered. After 48 h, cells were trypsinized, harvested, and counted (cell number average (mean) ± standard error measure (S.E.M.)). Three independent experiments were performed. The results (mean ± S.E.M.) are expressed as fold-change considering the control condition as 1.

### 2.3. Cell Cycle Fluorescence Activated Cell Sorter (FACS) Analysis

Cell cycle FACS analyses were carried out as described in [8]. Cells were cultured for 8, 12, 16, 24, and 48 h with or without U0126 in medium supplemented with 2% FBS. Cells were stained with propidium iodide (PI 50 μg/mL)/RNase (100 U/mL) (Sigma-Aldrich, cat. P4864 and R6513 respectively, St. Louis, MO, USA) for at least 3 h. Cells were then analyzed by Epics XL Flow Cytometer CyAn ADP (Beckman Coulter, Fullerton, CA, USA). and the results obtained were subjected to the FCS Express 5.1 software (De Novo, Los Angeles, CA, USA).

### 2.4. Cell Death Assay

Cells were cultured at a density of 5 × 10^5^ cells in 60 mm Petri dishes and starved. Cells were then treated with MEK1/2 inhibitor (U0126) at different concentrations (2.5, 5, 10, 15, and 20 µM) for 48 h. Cell death assays were performed as described in [8].

### 2.5. Chemotaxis Assay

Chemotaxis assays were performed as described in [8] by using cell culture inserts (12 well 8.0 µm pore size, Falcon, cat. 353182, Lincon Park, NJ, USA) posed in 12-well culture plates (Falcon, cat. 351143, Lincon Park, NJ, USA). Cell suspension without serum was added in the upper chamber of the trans-well (cell culture inserts (12 well 8.0 µm pore size, Falcon, cat. 353182, Lincon Park, NJ, USA), in DMEM alone, or in medium supplemented with U0126. The lower chambers were filled with DMEM alone (that represented our negative control of cell chemotaxis) or DMEM supplemented with HGF (that represented our positive control of chemotaxis). Cells were incubated for 16 h at 37 °C in an atmosphere containing 5% CO_2_ Cells, in the upper side of the filter, were mechanically removed, whereas the lower side of the filter, (with migrated cells), was fixed and stained with DAPI. Filters were studied by using a fluorescence microscope (Axioplan, Zeiss, Oberköchen, Germany). The results (mean ± S.E.M.) are expressed as fold-change (considering control condition arbitrarily as 1). Three independent experiments were performed.

### 2.6. Wound-Healing Assay (Collective Migration Assay)

Wound-healing assays were carried out by using double well culture inserts (Ibidi GmbH, Martinsried, Germany) as already described in [8]. Starved cells were detached and placed into the two wells of each insert allowing cell confluency. Then inserts were removed, and cells were cultured with 2% FBS DMEM (CTRL) or CTRL medium supplemented with HGF (40 ng/mL), U0126 (5 µM), U0126+ HGF. Each well was photographed by using a Zeiss Axiovert optical microscope (Zeiss, Oberkochen, Germany) equipped with a Nikon DS-Fi1 camera (Nikon Corporation, Tokyo, Japan). The cell free area was measured at time 0 (T0), immediately after insert removal, and after 24 h and 48 h of culture. The mean percentage of closed area, compared with the area empty of cells generated at T0, was calculated by ImageJ v 1.47 h software. Four independent experiments were performed.

Part of the samples cultured for 24 h after insert removal were also used for F-actin localization by using rhodamine phalloidin (Invitrogen Molecular Probes Eugene 1:40 dilution). TO-PRO3 iodide fluorescent dye 642/661 (1:5000 in PBS, Invitrogen, cat. T3605, Carlsbad, CA, USA) for nuclei staining was used, and samples were visualized by using a Leica confocal microscope (laser scanning TCS SP2 equipped with Kr/Ar and He/Ne lasers, Mannheim, Germany).

### 2.7. Immunofluorescence Analyses

Immunofluorescence analyses were performed to study the NT2D1 cytoskeleton changes analyzing vinculin and F-actin localization by using confocal microscopy analysis. The immunofluorescence protocol was reported in detail in [8]. Cells were fixed, permeabilized, and incubated overnight with anti-vinculin (Santa Cruz, cat. sc-73614; 1:50). After washings, cells were incubated with the FITC-conjugated donkey anti-mouse IgG secondary antibody (Jackson ImmunoResearch, cat. 715-095-150, dil. 1:200). TO-PRO3 iodide fluorescent dye 642/661 (1:5000 in PBS, Invitrogen, cat. T3605, Carlsbad, CA, USA) was used for nuclei staining. Rhodamine phalloidin (Invitrogen Molecular Probes Eugene 1:40 dilution) was used for F-actin detection. Samples were analyzed by using a Leica confocal microscope (laser scanning TCS SP2 equipped with Kr/Ar and He/Ne lasers, Mannheim, Germany).

The colocalization of vinculin and F-actin at the focal contacts was analyzed by Leica confocal software LAS-AF-Lite_2.6.3.

### 2.8. Western Blot Analyses

To better investigate the activation status of ERK, the cells were cultured for 5 min, 15 min in DMEM 2% FBS with or without HGF. On the other hand, pathway inhibition was investigated by treating cells with U0126 for 5, 15, 30, 60 min, 24, and 48 h. 

Times of 5 min, 24 h, and 48 h were used for HGF or U0126 treatments; to evaluate Snail, vimentin, E-and N-cadherin expression we used 24 and 48 h as time points. At the end of these culture times, cells were lysed in RIPA buffer containing protease and phosphatase inhibitors (Roche, cat. 04693124001 and 04906837001, Mannheim, Germany) and analyzed by Western blotting. The detailed protocol was reported in [8]. The membranes were incubated for 16 h at 4 °C with the following primary antibodies: anti-p44/42 MAPK (ERK1/2) (rabbit monoclonal, 137F5), and anti-phospho-p44/42 MAPK (ERK1/2) (rabbit monoclonal, Thr202/Tyr 204), antibodies both from Cell Signaling, Danvers, MA 1:1000. In addition, anti-Snail, anti-vimentin, anti-E-cadherin, anti-N-cadherin rabbit monoclonal antibodies were used (all from Cell Signaling, Danvers, MA, USA; 1:1000) (C15D3; D21H3; 24E10; D4R1H). The HRP conjugated secondary antibodies used in this study were: anti-rabbit or anti-mouse IgG (both 1:5000, GE Healthcare UK Limited, Buckinghamshire, England, cat. NA9340V). Protein normalization was performed using, monoclonal anti-α-actin antibody directly conjugated with HRP (1:10,000, Sigma-Aldrich, cat. A3854, St. Louis, MO, USA) or mouse-anti-GAPDH (SantaCruz Biotechnology 1:1000, Dallas, TX, USA). ECL Western blotting detection reagent, (Euroclone, cat. EMP011005, Pero, Italy) was used and filters were analyzed by ChemiDoc XRS. Acquired images were processed by Image Lab software (Bio-Rad Laboratories, Hercules, CA, USA). Phospho-ERK densitometric profiles were normalized versus total ERK/α-actin. At least three independent experiments were performed.

### 2.9. Scanning Electron Microscopy (SEM)

An amount of 9 × 10^4^ NT2D1 were cultured and treated with HGF and U0126 for 24 h. After that, samples were fixed in glutaraldehyde 2.5% overnight (O.N.). Sample preparation was described in detail in [5]. Supra 40 FESEM (Zeiss, Jena, Germany) (Department. of Experimental Medicine of the Università degli studi della Campania Luigi Vanvitelli) was used for sample observation and study.

### 2.10. Statistical Analyses

The data of this work are presented as the mean value ± standard error (SEM). Statistical significance was determined using two different software types: Sigmaplot 14 Data Analyzer Software and GraphPad Prism 8.0.2. Raw data were analyzed by using Student’s *t*-test and ANOVA tests (for multi-group comparison).

## 3. Results

### 3.1. ERK Is Constitutively Active in NT2D1 Cells, but HGF Administration Increases ERK Phosphorylation

It is well known that the HGF/c-MET system is able to activate ERK/MAPK signaling, even though no data are available so far concerning the activation of this pathway in NT2D1 cells. In order to evaluate whether c-MET-triggered ERK activation was mediated by HGF we evaluated by Western Blot phospho- and total p44/42 MAP kinase (ERK1/2) protein, both in basal condition and after HGF administration at different times (5, 15 min). Densitometric analysis of the bands revealed that ERK is phosphorylated in the control condition and that HGF significantly increased the phosphorylation of ERK in the activator site within a short time, starting from 5 min after treatment (HGF vs. CTRL *p* < 0.05) (Figure 1).

### 3.2. Pharmacological Approach to Block ERK Pathway

We pharmacologically inhibited the ERK activity by administering the MEK1/2 inhibitor U0126 in culture (Appendix A). We used this strategy to test the involvement of the ERK pathway in HGF-dependent and HGF-independent NT2D1 cell proliferation, collective migration, and chemotaxis. To identify the non-toxic dose of selective MEK1/2 inhibitor (U0126) in NT2D1 cells, we performed cell death flow cytometry analysis by culturing NT2D1 cells with different concentrations of the inhibitor (2.5, 5, 10, 15, and 20 µM) for 48 h (Section 2). Cell viability profile revealed that U0126 is not toxic on NT2D1 cells when administered at 2.5 and 5 µM (*p* = ns), but it shows a progressive dose-dependent toxicity starting from 10 µM to 20 µM as is evident from the graphical representation of the percentage of live cells with respect to the control condition (10 µM *p* ≤ 0.05; 20 µM *p* ≤ 0.01) (Figure 2A).

To assess inhibition efficacy, we performed Western Blot analyses of phospho- and total ERK1/2 from cells cultured in basal condition or with different U0126 doses (2.5 and 5 µM), which revealed that U0126 is able to decrease phosphorylated protein already at the 2.5 µM dose, even if the 5 µM dose shows more efficacy in its effect.

Indeed, 5 μM U0126, was chosen as the highest dose that could be used in culture to avoid toxic effects. This concentration of the inhibitor is able to statistically inhibit phospho-ERK over time as revealed by Western blot analysis of protein extracted from NT2D1 cells cultured in basal condition and with 5 µM U0126 treatment for 5, 15, 30, 60 min (Figure 2B,C).

### 3.3. UO126 Inhibits Both Constitutive and HGF-Dependent ERK1/2 Phosphorylation

To better clarify whether phospho-ERK 1/2 recruitment depends on the activation of the c-MET pathway, we performed Western Blot analyses of phospho- and total ERK1/2 on proteins extracted from NT2D1 cells cultured in basal condition or treated with HGF, or U0126, or U0126 + HGF for 5 min, 24 h, and 48 h. The densitometric analyses of the bands confirmed that ERK1/2 is actively recruited after HGF administration, and that the inhibitor is able to reduce its activation in a statistically significant way up until 24 h (Figure 3).

More in detail, results clearly show the expected significant increase in the p-ERK/ERK ratio in HGF-treated samples with respect to the control condition, indicating activation of the ERK dependent pathway after 5 min and up until 24 h after administration (HGF vs. CTRL *p* < 0.001). The dose of 5 µM U0126 in combination with HGF was able to revert this cellular response (U0 + HGF vs. HGF *p* < 0.001). Moreover, U0126 administration alone affected the endogenous phosphorylation with respect to the control condition (U0126 vs. CTRL *p* < 0.001) (Figure 3). Control values were considered conventionally as 1, and sample values were calculated as fold change with respect to control values. These results indicate that the ERK phosphorylation, both endogenous and HGF-triggered, is inhibited by U0126 administration.

Although a trend in U0126 inhibitory capability is maintained up to 48 h, HGF administration did not increase the p-ERK/ERK ratio after 48 h of culture, and this is in line with results previously obtained, in which a gradual decrease in c-MET availability was detected in culture after HGF administration, [4] because of receptor turnover.

### 3.4. HGF-Induced NT2D1 Cell Proliferation, Is Mediated by MAPK/ERK Pathway

We performed proliferation experiments, as described in the Section 2, and as already described in previous papers [4,5,8]. The increase of HGF-triggered cell proliferation is evident only after 48 h of culture after HGF stimulation, because of the relatively slow duplication rate of this cell line [4]. Therefore, for proliferation assay purposes we cultured NT2D1 cells for 48 h in control conditions (with or without HGF) or in the presence of U0126, alone or in combination with HGF. As expected, HGF administration induced a significant increase of cell number compared with control samples (total cell number: 29.7 × 10^4^ ± 0.48 vs. 19.9 × 10^4^ ± 0.69 respectively, *p* < 0.001). Notably, we observed that the treatment with HGF + U0126 completely abrogates the HGF-induced NT2D1 cell proliferation (total cell number: 13.6 × 10^4^ ± 0.66 vs. 29.7 × 10^4^ ± 0.48 respectively *p* < 0.001), indicating that ERK has a role in NT2D1 cell proliferation induced by HGF. Notably, the decrease in total cell number using inhibitor alone compared with the control samples, indicates that ERK is involved also in endogenous and HGF-independent NT2D1 cell proliferation (total cell number: 13.9 × 10^4^ ± 0.74 vs. 19.9 × 10^4^ ± 0.69 respectively; *p* < 0.001) (Figure 4A).

To better characterize this phenomenon, cell cycle analyses were performed. As expected, starved NT2D1 cells (maintained for 15 h without serum (T0)) showed an increase of G1-phase cell percentage. Cell cycle FACS analyses were then performed after 8 h, 12 h, 16 h, and 24 h of culture, both in the control condition (2% FBS) and after U0126 (5 µM) administration. Analyses revealed an increase of G1 phase cell percentage at 8, 12, and 16 h in U0126 treated cells with respect to control samples (U0126 vs. CTRL *p* < 0.05). After 24 h no modulation in cell cycle phases was observed (*p* = ns) (Figure 4B). The same observation was obtained at 48 h (Appendix A). Taken together these results demonstrate that ERK is directly recruited in the c-MET signaling activated by HGF and is also involved in the HGF-independent cell cycle progression, since it is able to modulate the G1 phase of NT2D1 cell cycle causing a slight cell cycle slowdown when administered alone.

### 3.5. Inhibition of the ERK Pathway Does Not Result in Inhibition of Cell Motility

We established that HGF acts as a chemo-attractant for NT2D1 in Boyden chamber migration assays [4]. We also demonstrated that this migration is c-MET specific and that the PI3K/AKT and c-Src pathways are both involved in HGF-dependent NT2D1 chemotaxis [5,8]. We wondered whether even ERK pathways could drive motile functions of NT2D1. Therefore, we performed migration experiments using the already mentioned MEK1/2 inhibitor, U0126. As shown in (Figure 5A,B), migration significantly increases in the presence of HGF with respect to the control condition (* *p* < 0.05), as expected.

Indeed, the treatment of cells with U0126 does not significantly reduce the migratory effect induced by HGF (*p* = ns). On the contrary, the inhibition of ERK, seems to encourage HGF-independent NT2D1 cell chemotaxis with respect to the control condition even if values obtained are not statistically significant with respect to control values. (Figure 5B).

In chemotaxis experiments, cells migrate in a polarized way as a single cell after an epithelial to mesenchymal transition (EMT). However, this is not the only way for cell movement. We studied the role of the ERK pathway on NT2D1 cell motility even by using a collective migration assay (wound healing assay) in which the cells move, maintaining cell–cell interactions, coordinating their cytoskeletal dynamics and intracellular signaling. We performed wound healing assay, as described in the Section 2, in the following experimental conditions: CTRL, HGF, U0126 alone, or in combination with HGF; then we measured the open area of the sample at time 0 (T0, considered as 100% of the open area) and after 24 and 48 h of culture from the wound induction. Then we compared the mean percentage of closed area with respect to the empty area at the T0 condition. As expected, HGF administration increases collective cell migration with respect to the control condition (HGF vs. CTRL *p* < 0.001) after 48 h of culture. However, the percentage of closed area in samples treated with U0126 alone or U0126 in combination with HGF showed values similar to HGF condition and, therefore, significantly higher compared to control condition (U0 + HGF and U0126 vs. CTRL *p* < 0.001), demonstrating that ERK inhibition alone does not impair constitutive and HGF-induced collective migration, but surprisingly encourages it even when U0126 is administered alone (Figure 5C,D). To better describe this phenomenon, we studied by confocal microscopy the F-actin distribution pattern of migrating cells, 24 h after wound induction, finding that in all the treated samples cells migrate both in a collective way, maintaining the epithelial phenotype of the cells, and as single cells, detaching from the cleading edge of the cell migration front (Figure 5E).

### 3.6. HGF and U0126 Treated Cells Show Different Cell Morphology

Since U0126 acts on NT2D1 cells, promoting a migratory behavior independently from HGF administration, we wondered if this inhibitor is also able to significantly influence cell morphology in the acquisition of a “mesenchymal-like phenotype”. To this aim, cells were treated for 24 h with HGF, or U0126 or both compounds and analyzed by scanning electron microscopy (SEM). This analysis revealed, as expected, that HGF stimulation significantly modifies cell shape: indeed, cells appeared stretched and enriched in membrane protrusions, microvilli-like structures, and membrane ruffles formation. This experimental condition also determines the formation of filopodia and lamellipodia, confirming the migratory attitude induced by HGF administration in this cell line (Figure 6A).

On the contrary, control cells have a smooth membrane surface and membrane activity appears less evident. In U0126 treated cells and when U0126 is co-administered with HGF membrane, activity is stimulated, inducing the appearance of filopodia and membrane ruffles similar to that observed after HGF administration (Figure 6A). This highlights the relevance of ERK inhibition in modulating NT2D1 cell behavior accordingly with the results that we obtained in the wound-healing assay.

To better investigate cell morphology and cytoskeleton reorganization during migration, confocal microscopy analysis of F-actin and vinculin was performed in wound-healing assay samples after 24 h of culture at the migration front, evaluating focal adhesion complexes (focal contacts) distribution pattern (Figure 6B,C). This time point was chosen because at 24 h, the wound is not completely closed, and therefore the morphology of the cell leading edge is observable. We observed both in control and HGF-treated cells the presence of cortical cytoskeletal actin and stress fibers, which are features of cell adhesion and possible collective migration capability. Notably, in HGF treated samples, actin-based lamellipodia and filopodia formation are more frequent than in control samples. Interestingly in all the U0126 treated cells the stress fibers together with filopodia and lamellipodia are formed, in a condition much more “HGF-like” and therefore not similar to control cells (Figure 6B). The colocalization analysis of vinculin an F-actin revealed a good colocalization of these markers at focal contact level. Notably, focal contacts appear more numerous in control samples then in treated ones. Moreover, in control samples we observed that focal contacts are mainly distributed in the central part of the cell, by which means cells maintain a stable contact with the substrate. Focal contacts are instead mainly distributed at the cell membrane leading edge in all treated samples (Figure 6C).

### 3.7. U0126 Treatment Increases p-AKT Level in NT2D1 Cell Line

In our previous study we reported that PI3K/AKT pathway is involved in the positive regulation of NT2D1 migration and invasion [5]. Based on this knowledge and trying to understand the reason why ERK inhibition exerted positive control of NT2D1 cell migration, we wondered whether ERK inhibition could result in the activation of compensatory pathways such as PI3K/AKT. As revealed by Western blot analyses performed on NT2D1 cells treated with U0126 alone or with HGF, we observed that ERK inhibition stimulates the activation of the AKT pathway after 15 min (U0126 and HGF vs. CTRL *p* < 0.01) (Figure 7A,B) triggering AKT phosphorylation.

### 3.8. HGF and U0126 Treated Cells Show Different Modulation of EMT Markers

An epithelial–mesenchymal transition (EMT) is a process that biochemically enables the epithelial cell to lose its junctional contacts and assume a mesenchymal cell phenotype: this implies enhanced migratory capacity, invasiveness, elevated resistance to apoptosis, and greatly increased production of ECM components. In many cases, the factors involved in EMT are also used as biomarkers to demonstrate the passage of a cell through an EMT: for instance, the increase of Snail, N-cadherin and vimentin expression levels, are considered markers of mesenchymal cell phenotype acquisition, together with the decrease of E-cadherin expression marker.

Therefore, based on results described in the previous paragraph, we performed Western blot analyses for some relevant proteins involved in the EMT switch, such as Snail, vimentin, N- and E-cadherin on NT2D1 cells cultured for 24 h and 48 h in basal condition and after HGF, UO126, U0126 + HGF (Figure 8).

Densitometric analysis of the bands revealed that both HGF and U0126 treatments significantly increase Snail protein level with respect to the control condition after 24 h (HGF vs. CTRL *p* < 0.01; U0126 vs. CTRL *p* < 0.05). U0126 when administered with HGF restores Snail protein level to control condition (*p* = ns), significantly reverting single treatments effects (U0126 + HGF vs. HGF, *p* < 0.01; U0126 + HGF vs. U0126 *p* < 0.05) (Figure 8A). At 48 h no modulation in Snail protein level after HGF treatment was observed. Notably, at the same culture time, U0126 administration alone reduces its expression with respect to control samples (*p* < 0.05) (Figure 8B). As far as vimentin expression is concerned, no modulation after HGF treatment was observed with respect to control. U0126-treated cells increase vimentin expression with respect to the control condition (U0126 vs. CTRL *p* < 0.05) and this difference persists after 48 h of culture (U0126 vs. CTRL *p* < 0.01) (Figure 8C,D). Notably, in this case, at 24 h HGF + U0126 treatment does not restore the control condition, vimentin levels being significantly higher with respect to untreated cells (U0126 + HGF vs. CTRL *p* < 0.001) (Figure 8C). However, after 48 h of treatment, vimentin protein levels return similar to the control condition (U0126 + HGF vs. CTRL *p* = ns), and significantly lower with respect to single agent treatment (U0 + HGF vs. HGF and U0 + HGF vs. U0126, *p* < 0.01 and *p* < 0.001 respectively) (Figure 8D). Considering the cadherins switch (E-cadherin vs. N-cadherin), Western blot analyses revealed that after 24 h of treatment, N-cadherin reflects vimentin expression: U0126 and HGF + U0126 treated cells increase protein expression with respect to the control condition (U0126 and U0 + HGF vs. CTRL *p* < 0.05) (Figure 8E). At 48 h protein expression decreases and U0 + HGF treatment significantly reduces N-cadherin expression with respect to the control condition (U0 + HGF vs. CTRL *p* < 0.01). (Figure 8F). Notably, HGF administered alone does not promote evident changes in N-cadherin expression, but it is able to increase E-cadherin after 24 h (HGF vs. CTRL *p* < 0.05). Moreover, U0126 and HGF + U0126 treated cells also increase E-cadherin expression with respect to the control condition (U0126 and U0 + HGF vs. CTRL, respectively *p* < 0.001 and *p* < 0.01) (Figure 8G). Finally, after 48 h U0 + HGF treatment significantly reduces E-cadherin expression with respect to the control condition (U0 + HGF vs. CTRL *p* < 0.01) (Figure 8H).

## 4. Discussion

The term “testicular germ cell tumors (TGCTs)” defines a group of neoplasms whose incidence has been continuously increasing among young men. Despite a good prognosis and without adjuvant treatments, a percentage of non-seminomatous and seminomatous lesions relapse and patients still develop chemo-resistance and die of tumor progression [21,22,23]. Notably, adjuvant chemotherapy often leads to detrimental platinum-associated side effects and this evidence acquires worth considering the young age of the patients at the onset of the disease. For these reasons, in the last decade, the study of alternative therapeutic strategies to achieve the goal of minimising therapy-related toxicity has been improving, providing long term clinical outcome [24,25]. However, the exploration of possible second-generation therapies in pre-clinical studies is still limited and has given only frustrating results [26]. Notably, the genetic signature of these oncological lesions is still not known, and it is commonly accepted that these malignancies are the consequence of the combination of environmental and (epi)genetic alterations. A gain has been reported of copies of chromosome p12 in these cancers in fact [27], but this observation has failed to find driver genes that could be used for target therapies. As a matter of fact, TGCT patients that relapse have not the possibility to access personalized therapies, that are instead available for most other malignancies [28].

In the last decade, the investigation of TGCTs molecular features has had pulse in identifying novel therapeutic targets and discovering more effective and less toxic therapies to personalise the cure of these neoplasms [29,30]. Our research group studied c-MET/HGF expression in type II TGCT lesions and cell lines. Some of our previous studies investigated the c-MET/HGF system involvement in non-seminoma cell malignant behavior [4,5,8]. The role of this system in the progression of several solid oncological lesions is well known [31], and its signal transduction pathway influences the proliferation and survival of cancers driven by growth factor receptors. In the present paper, we studied the role of ERK adaptor in the regulation of HGF-dependent and HGF-independent NT2D1 cell proliferation and migration. Our results clearly indicate a role of ERK pathway both in basal and HGF-triggered NT2D1 cell proliferation. Intriguingly, we obtained similar results using Src Inhibitor-1 alone, as reported in our previous work [8]. This indicates a redundant role of c-Src and ERK in the regulation of this biological response. It is tempting to speculate that in those basal culture conditions, independently from c-MET pathway activation, ERK could be used by other constitutively activated pathways that are responsible for the activation of the cell cycle.

Since we previously demonstrated that HGF-dependent c-MET activation can induce migrating phenotype in NT2D1 cells as well as the involvement of PI3K and Src pathways in this phenomenon [5,8] we decided to evaluate if ERK pathway could affect even this aspect of NT2D1 cell behavior. The results reported in chemotaxis and collective migration assays indicated that the inhibition of ERK does not inhibit HGF-triggered cell migration, but conversely, U0126 given alone promotes cell migration. These observations are in line with the reported SEM analysis by which a clear change of cell morphology to a “migratory” phenotype is observable after U0126 administration. Even the analysis of focal contact distribution revealed that these structures are mainly localized at the cell leading edge of HGF and U0126 treated samples, confirming the “migratory” phenotype acquired by the cells. Notably, in wound healing assays it has been observed that in U0126 and HGF treated samples part of the cells migrate as single cells.

Taken together these results let us speculate that NT2D1 cells could undergo, at least partly, epithelial–mesenchymal transition (EMT) and therefore we decided to investigate the modulation of EMT markers in the different cultural conditions. The results obtained studying epithelial mesenchymal transition markers gave information compatible with the acquisition of an “intermediate stage” of the EMT process both after HGF or U0126 treatments [32]. In particular, the increase in mesenchymal phenotype observed in NT2D1 cells after HGF administration is in line with the concomitant increase of Snail protein after this treatment and could at least explain their migrating capability and at the same time the attitude in promoting collective migration. Simultaneously, the increases of vimentin, N-cadherin, Snail, in U0126 treated samples are in line with the acquisition of a “mesenchymal” phenotype when ERK signal is inhibited.

Recently, it has been proposed that the partial EMT (pEMT) of the “intermediate stage” is evidence that underlines bi-directional crosstalk between tumor cells and the surrounding microenvironment in the acquisition of pEMT phenotype. Even if we still do not know the mechanism outlining the acquisition of pEMT phenotype, it is becoming increasingly evident that the tumor microenvironment contributes significantly to the acquisition of this phenotype and that, even for this reason, this pEMT is dynamically reversible [33]. The concomitant activation of AKT protein, as compensatory pathway, after U0126 treatment, reinforces this hypothesis and sheds light to a double inhibition strategy, when single agent therapy promotes changes in cell signaling that potentiate expression of the mesenchymal phenotype [34]. Notably, the observation that U0126 does not increase the amount of HGF-triggered pAKT allows us to speculate that ERK and pAKT pathways represent separated and independent pathways when activated by the c-MET tyrosine kinase multifunctional docking site. Conversely, the constitutively active ERK pathway in NT2D1 cells is in balance with the PI3K/AKT pathway and therefore, in our hypothesis, the inhibition of ERK resulted in a compensatory activation of the AKT pathway. The cross-activation between ERK and AKT pathways has, in fact, already been reported in the literature because of single agent treatments; the interaction between the parallel pathways may explain the poor capability of agents that individually target them. Single-agent inhibition has been shown to lead to compensatory up-regulation of ERK or AKT activity in models of glioblastoma [13]. Finally, it has been emerging that acquired pharmacological resistance or changes in cell signaling in response to single molecule treatment, may potentiate acquisition of the mesenchymal phenotype, as occurs for example in melanoma [34]. Taken together these results can explain, at least in part, the frustrating results obtained by single drug administration in the treatment of these oncological lesions and strongly indicate the necessity of a combined therapy to possibly address the clinical challenge of lesions resistant to conventional therapeutic protocols.

## 5. Conclusions

The data presented in this work demonstrate that ERK is constitutively activated in NT2D1 cells and is involved in the control of cell proliferation. Moreover, our observations indicate that ERK inhibition may result in the activation of compensatory pathways (i.e., PI3K/AKT). This observation indicates that the inhibition of single onco-adaptor protein could exert paradoxical effects that promotes malignant behavior of cancer cells instead of controlling malignancy. Therefore, the results reported herein highlight how the molecular feature and drug sensitivity of each oncological lesion should be investigated in depth before setting-up the therapy. This stimulates further investigations and the development of cellular in vitro models, derived from patient lesions, that could represent a good tool allowing prediction of responders and not responders for TGCTs personalized target therapies.

## Figures and Tables

**Figure 1 biomedicines-11-01894-f001:**
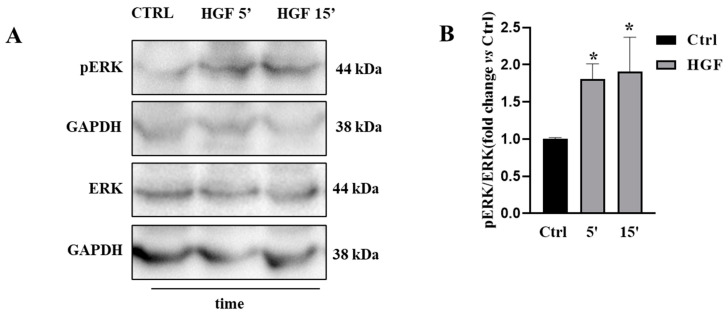
(**A**) Western Blot analyses of phospho- and total p44/42 MAP kinase (Erk1/Erk2) were performed on cells cultured in basal condition, 5, and 15 min after HGF administration (40 ng/mL). (**B**) Graphical representation of densitometric analyses of the bands (* vs. CTRL *p* < 0.05). Results are expressed in fold change, with the control considered as 1 (± standard error of the mean (S.E.M.)). Four independent experiments were performed.

**Figure 2 biomedicines-11-01894-f002:**
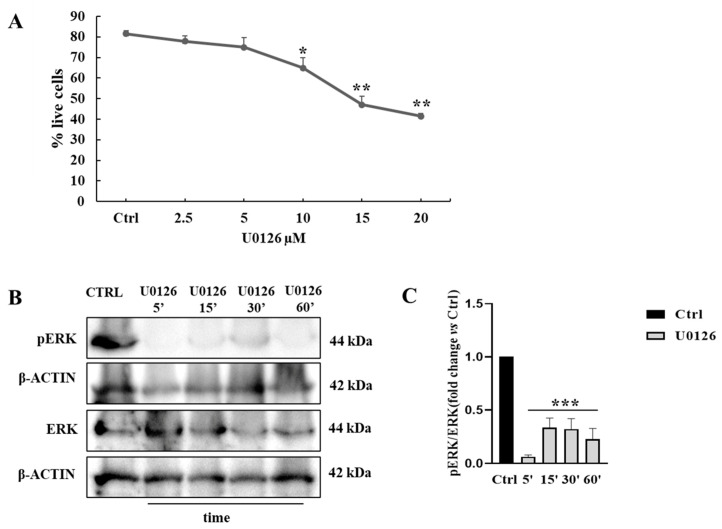
(**A**) Cell death flow cytometry analysis on NT2D1 cells. Graphical representation of the percentage of live cells (fold change) obtained by culturing NT2D1 cell in control conditions and U0126 doses (2.5, 5, 10, 15, and 20 µM) (Mean ± S.E. is reported) (* *p* ≤ 0.05; ** *p* ≤ 0.01). (**B**) Western Blot analyses of phospho- and total ERK1/2 performed on cells cultured in basal condition and 5, 15, 30, 60 min after U0126 administration. (**C**) Graphical representation of densitometric analyses of the bands (5 µM) *** vs. CTRL *p* ≤ 0.001. Results are expressed in fold change, with the control considered as 1 (± S.E.M.)). Four independent experiments were performed.

**Figure 3 biomedicines-11-01894-f003:**
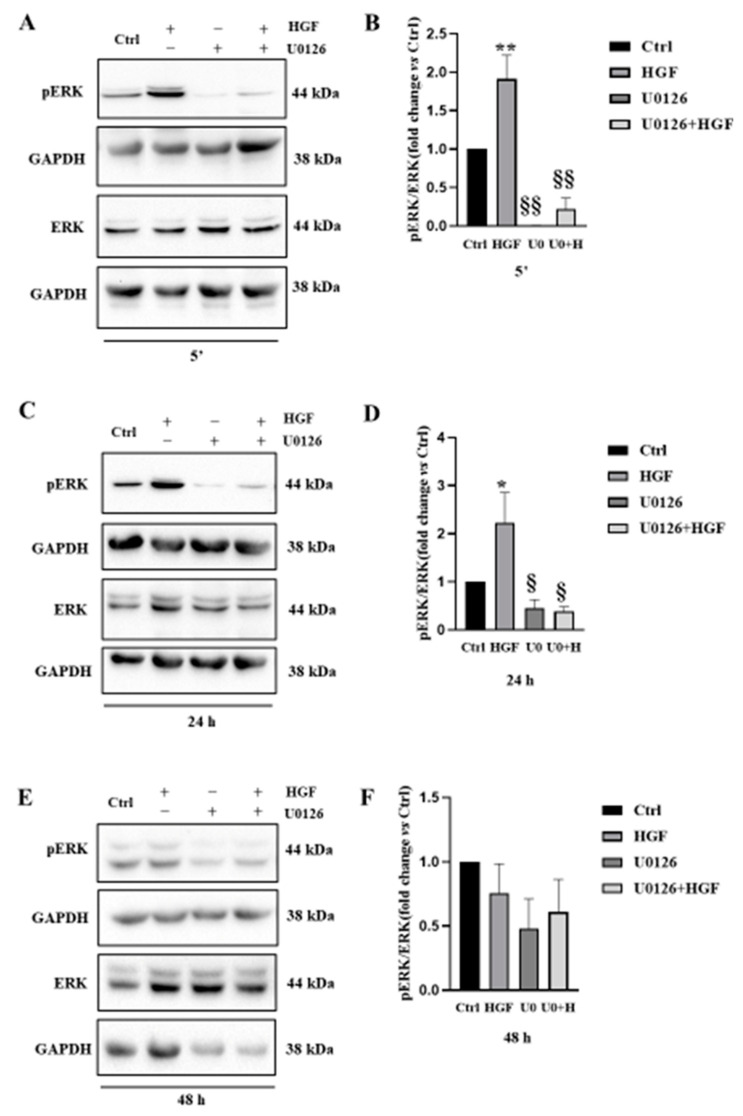
(**A**,**C**,**E**) Western Blot analyses of phospho- and total ERK1/2 performed on cells cultured in basal condition and with HGF, U0126, U0126 + HGF for 5 min, 24 and 48 h. (**B**,**D**,**F**) Densitometric analyses of the bands are reported on the right (§§ vs. ** vs. Ctrl *p* < 0.01; § vs. * vs. Ctrl *p* < 0.05). Results are expressed in fold change, with the control considered as 1 (± standard error of the mean (S.E.M.)). Three independent experiments were performed.

**Figure 4 biomedicines-11-01894-f004:**
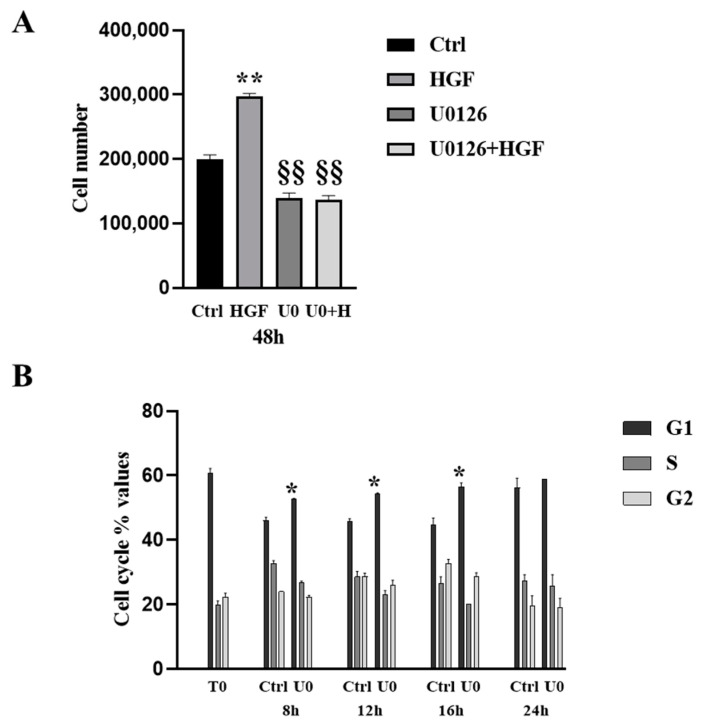
(**A**) NT2D1 cell count after 48 h of culture in control condition (CTRL), or in medium supplemented with 40 ng/mL HGF, or 5 μM U0126, or the combination of both treatments (U0126 + HGF). As expected HGF administration causes a significant increase of cell number (** vs. CTRL *p* < 0.001). Conversely, U0126 alone or in combination with HGF triggers a significant reduction in cell number not only with respect to HGF treated samples (** vs. §§ *p* < 0.001), but also compared to control samples (CTRL vs. §§ *p* < 0.001). The experiment was repeated three times, and each experiment performed in triplicate. We reported in the graph the mean values ± S.E.M. (**B**) Cell cycle analysis of NT2D1 cells cultured for 8, 12, 16, 24 h in medium supplemented or not supplemented with U0126. The graph illustrated the mean percentage of cells for each phase of cell cycle ± S.E. We reported a significant increase of the percentage of G1 phase NT2D1 cells after 8 h, 12, 16 h U0126 administration (* *p* < 0.05). The experiment was repeated three times, and each experiment performed in triplicate.

**Figure 5 biomedicines-11-01894-f005:**
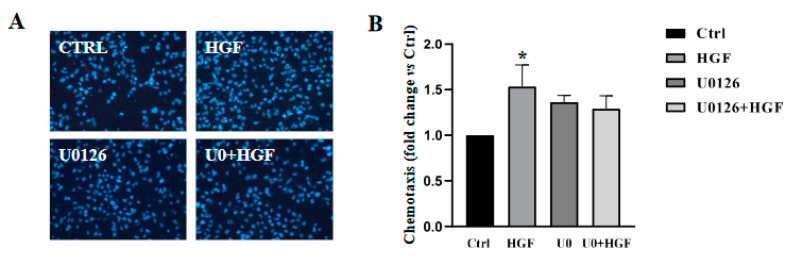
(**A**,**B**): Effect of U0126 on cell NT2D1 cell migration (chemotaxis). (**A**) representative images of NT2D1 cell migration. Images were recorded at 20× magnification. At least three independent experiments were performed. (**B**) Quantitative analysis of chemo attracted NT2D1 cells. The values were calculated as “fold change” (±S.E.M.) compared to the control, which was considered as 1. U0126 in combination with HGF does not abrogate the migratory effect induced by HGF (* *p* < 0.05). (**C**–**E**): Effect of U0126 on NT2D1 cell collective migration. (**C**) Representative images recovered immediately after insert removal (T0) 24 and 48 h after wounding. (Scale bar: 100 µm). (**D**) Quantitative analysis of wound closure after 48 h. Data are expressed as the mean percentage of closed area compared with the respective T0 condition (** *p* < 0.001). U0126 was not able to inhibit the collective migration of the cells cultured for 24 h and 48 h in presence of HGF (*p* = ns). Five independent experiments were performed. (**E**) Representative images of F-actin detection (**red signal**) in wound healing assay samples fixed after 24 h of culture. It is evident that the number of cells migrating as single cells is increased in treated samples (Scale bar: 100 µm).

**Figure 6 biomedicines-11-01894-f006:**
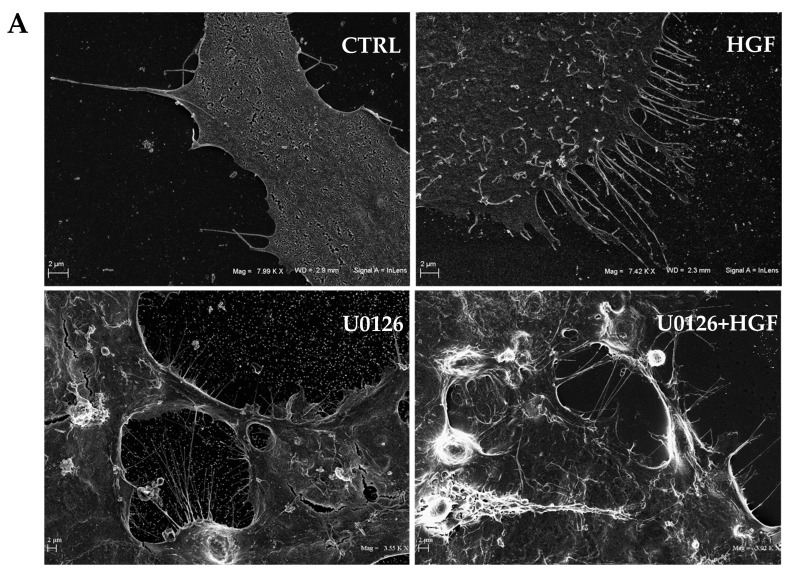
(**A**) Scanning electron microscopy analysis. Representative images of NT2D1 cells cultured for 24 h in the control condition, or treated with HGF, U0126, or their combination. Scale bar 2 µm. (**B**) Representative confocal images of F-actin and vinculin organisation in NT2D1 cells during wound healing experiment, recovered at 24 h after the wound. Images were recovered at the leading edge of the wound: vinculin (**green signal**); F-actin (**red signal**); the merging picture is provided on the right side of the panel. Scale bar 37.5 µm. Three independent experiments were performed. (**C**) Colocalization analysis of vinculin (**green**) and F-actin (**red**) of the merging images reported in the panel (**B**). In the X/Y axes, at the level of the dotted lines, the colocalization (**green and red picks**) is shown. The analysis shows a colocalization of vinculin (**green**) and F-actin (**red**) peaks revealing also that this colocalization is mainly distributed in the central part of the cells in control samples, whereas it appears mostly displaced at the free front of cell membrane in treated cells.

**Figure 7 biomedicines-11-01894-f007:**
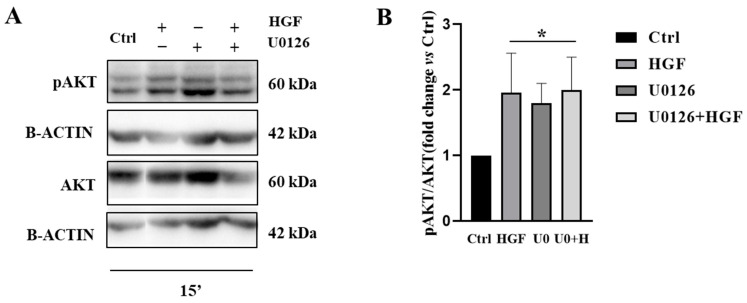
(**A**) Western Blot analyses of phospho- and total AKT performed on cells cultured in basal condition and with HGF, U0126, U0126 + HGF for 15 min. Densitometric analyses of the bands are reported (**B**) (* vs. CTRL *p* < 0.01). Results were expressed in fold change, with the control considered as 1 (±standard error of the mean (S.E.M.)). Three independent experiments were performed.

**Figure 8 biomedicines-11-01894-f008:**
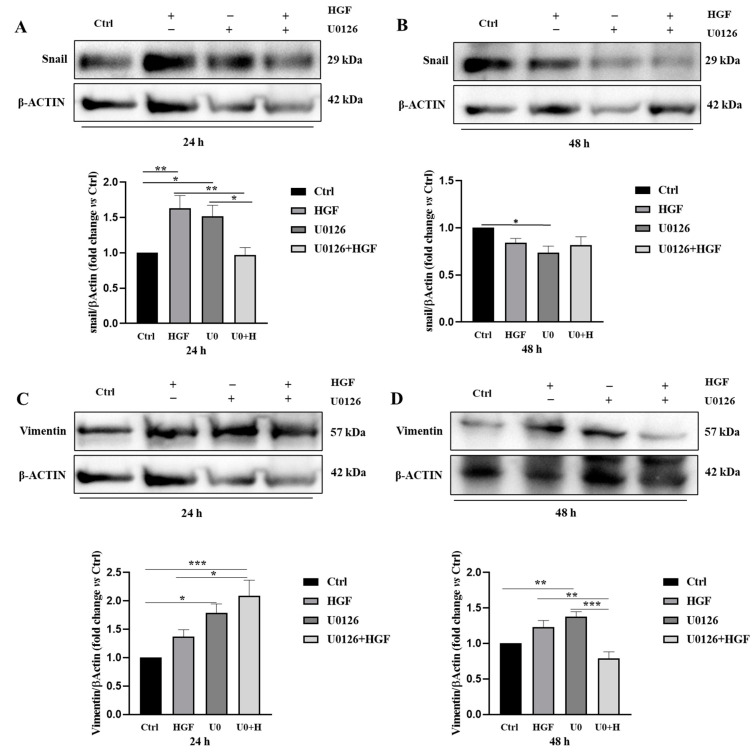
Single treatments with HGF and U0126 induce a modulation of both epithelial and mesenchymal markers. U0126 + HGF treatment does not always restore the expression level to control values. (**A**,**C**,**E**,**G**) Western blot analyses of Snail, vimentin E-cadherin and N-cadherin in NT2D1 cell lines cultured in basal conditions (CTRL), with 5 µM U0126, with 40 ng/mL HGF, and with U0126 + HGF at 24 and 48 h (**B**,**D**,**F**,**H**). The densitometric analysis of the bands are reported for each protein (* *p* < 0.05; ** *p* < 0.01; *** *p* < 0.001). Results are expressed in fold change, with the control considered as 1 (± standard error of the mean (S.E.M.)). Four independent experiments were performed.

## Data Availability

Data supporting reported results can be found on request from the corresponding author. The whole Western blots are reported in a Appendix A.

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
