# Peer review of "ERK Signaling Pathway Is Constitutively Active in NT2D1 Non-Seminoma Cells and Its Inhibition Impairs Basal and HGF-Activated Cell Proliferation"

_biomedicines, 2023, doi:10.3390/biomedicines11071894_

Round 1
Reviewer 1 Report
In this study, the authors investigated the role of MAPK/ERK pathways in the HGF-dependent and HGF-independent NT2D1 cells biological responses. The authors concluded that the use of single inhibitor of ERK pathway, resulted in activation of correlative pathways (i.e. AKT), and together with the reported higher level of HGF in EC patients highlighted in their previous work, indicating that the inhibition of single onco-adaptor protein could exert paradoxical effects.
Comments
The reviewer has some concerns as follows:
1. The title of this manuscript that “ERK signalling pathway is constitutively activated in NT2D1 non-seminoma cells and modulated by c-MET/HGF axis” should be revised. In this study, there is no any evidence to support that the ERK signaling pathway can be modulated by cMET/HGT axis in NT2D1 cells.
2. One of the major concerns is that only one non-seminoma cell line (NT2D1) and one pharmacological inhibitor for ERK pathway (U0126) were used. The overall evidence is quite weak. Only using a cancer cell line is not convincingly effective on this cancer. In addition, a pharmacological inhibitor by itself affects cell functions such as growth or migration in NT2D1 cells. How to prove that it would really interfere with cell function or affect the effect of HGF on cells by specifically inhibiting the ERK signal pathway. U0126 may have pharmacological effects other than affecting ERK signal pathway. Specifically, the more rigorous designs for cancer cell experiments should include two different cell lines and use methods such as siRNA that can directly block gene regulation of ERK signaling pathway.
3. The blot images in Figures 1 and 2 are unconvincing and do not match the statistical results. These data presentations cannot be accepted.
4. The migration images in Figure 5C- 48 h are confusing and unconvincing. This data presentation cannot be accepted.
5. In Figure 7, why U0126 did not affect the HGF-increased AKT phosphorylation? In Figure 1A, it seems that the obvious reduction of total protein for AKT leads to a proportional increase for p-AKT/AKT in HGF+U0126 group.
6. In Figure 8, the results are confusing and unconvincing. How to explain the results in HGF+U0126 group? Moreover, the hallmark of EMT is the upregulation of N-cadherin followed by the downregulation of E-cadherin. However, in Figure 8-E-H, the results for N- and E-cadherin are the opposite.
7. In general, the overall evidence is too weak to support the conclusions.
Author Response
Answer to Reviewer 1
Comments and query
In this study, the authors investigated the role of MAPK/ERK pathways in the HGF-dependent and HGF-independent NT2D1 cells biological responses. The authors concluded that the use of single inhibitor of ERK pathway, resulted in activation of correlative pathways (i.e. AKT), and together with the reported higher level of HGF in EC patients highlighted in their previous work, indicating that the inhibition of single onco-adaptor protein could exert paradoxical effects.
Comments
The reviewer has some concerns as follows:
- The title of this manuscript that “ERK signalling pathway is constitutively activated in NT2D1 non-seminoma cells and modulated by c-MET/HGF axis” should be revised. In this study, there is no any evidence to support that the ERK signaling pathway can be modulated by cMET/HGT axis in NT2D1 cells.
ANSWER 1: In this work we reported that HGF stimulation increases ERK phosphorylation, and therefore we assumed that c-Met/HGF system can modulate ERK signalling in this cell line. However, we agree that the statement of the title needs to be revised and to address the request of the reviewer we changed the title as follow: “ERK signalling pathway is constitutively activated in NT2D1 non-seminoma cells and its inhibition impairs basal and HGF-activated NT2D1 cell proliferation.
- One of the major concerns is that only one non-seminoma cell line (NT2D1) and one pharmacological inhibitor for ERK pathway (U0126) were used. The overall evidence is quite weak. Only using a cancer cell line is not convincingly effective on this cancer. In addition, a pharmacological inhibitor by itself affects cell functions such as growth or migration in NT2D1 cells. How to prove that it would really interfere with cell function or affect the effect of HGF on cells by specifically inhibiting the ERK signal pathway. U0126 may have pharmacological effects other than affecting ERK signal pathway. Specifically, the more rigorous designs for cancer cell experiments should include two different cell lines and use methods such as siRNA that can directly block gene regulation of ERK signaling pathway.
ANSWER 2: In our first study of c-Met/HGF axis in testicular germ cell tumours (Corano Scheri et al 2018) we reported our observations on different TGCT-derived cell lines observing that NT2D1 non-seminoma cells were the ones that have the best biological response to HGF in terms of proliferation, and migration. This is the reason why we decided to focus our study of the Tyrosine Kinase signalling pathway on NT2D1 cells, neglecting, in this study the other TGCT-derived cell lines. Currently, we are exploring in parallel the behaviour and pathways of seminoma cells, but the results obtained indicate that the seminoma cells respond to tyrosine kinase receptors activating completely different biological responses, and therefore the results on other TGCT-derived cell lines seem not comparable with the result reported in this manuscript.
Even if the RNA interference is a good way to decrease ERK availability the strategy is not superimposable with the enzymatic inhibition, that is so far the best method to impair kinase activity. As far as we know the use of U0126 as pharmacological inhibitor of ERK/MEK pathway is still used in several manuscript as single inhibitor.
- The blot images in Figures 1 and 2 are unconvincing and do not match the statistical results. These data presentations cannot be accepted.
ANSWER 3: We agree with the reviewer, and we repeated the western blot analyses. The blots of figure 1 and 2 has been changed with clearer ones.
- The migration images in Figure 5C- 48 h are confusing and unconvincing. This data presentation cannot be accepted.
ANSWER 4: We wish to thank the reviewer for the careful revision. The migration test has been repeated another time and some of the images have been changed with clearer ones. In addition, confocal analysis of F-actin of migrating cells has been added. To give a more immediate indication of migration capability of NT2D1 cells, the graph has been modified: on the Y axis we indicated the percentage of closed area, instead of the percentage of open residual area.
- In Figure 7, why U0126 did not affect the HGF-increased AKT phosphorylation? In Figure 1A, it seems that the obvious reduction of total protein for AKT leads to a proportional increase for p-AKT/AKT in HGF+U0126 group.
ANSWER 5: In our opinion HGF-triggered ERK and pAKT pathways represent separated and independent pathways recruited by the multifunctional docking site of c-Met receptor. Conversely, the constitutively active ERK pathway is in balance with PI3K/AKT pathway and therefore, in our hypothesis, the inhibition of ERK resulted in a compensatory activation of pAKT in and HGF-independent way. We added this hypothesis in the text of the discussion section.
Figure1 A does not report AKT quantification. The figure 7 A reported AKT and pAKT quantification, but, even if from the bands presented total AKT amount seem to decrease in HGF+U0126, analyzing all the western blots performed we did not find a significant reduction in total AKT level in HGF+U0126 samples.
- In Figure 8, the results are confusing and unconvincing. How to explain the results in HGF+U0126 group? Moreover, the hallmark of EMT is the upregulation of N-cadherin followed by the downregulation of E-cadherin. However, in Figure 8-E-H, the results for N- and E-cadherin are the opposite.
ANSWER 6: as stated in the work we do not find a complete EMT, but a partial EMT, that describes a quite instable state of the cells. The E-Cadherin increase is in line with the reported increase of collective migration in which part of the cells migrate as a monolayer maintaining and, possibly, reinforcing their junctional contacts.
- In general, the overall evidence is too weak to support the conclusions.
ANSWER 7: the Conclusion section has been thoroughly revised.
Reviewer 2 Report
This is amazing work. The article uses modern methods. All results are presented in an excellent way, the voluminous supplement contributes to the understanding of the work done. 1) In view of the large amount of material received, the authors must present the results of the work in the form of a graphical figure.
I have done additional peer review of the article as requested. I am sure that this work is worthy of publication. 1. The study "ERK signaling pathway is constitutively activated in NT2D1 non-seminoma cells and modulated by c-MET/HGF axis" is devoted to the regulatory role of this signaling cascade in the pathophysiology of cancer cells. The authors used one cell line (NT2D1 embryonal carcinoma cells). However, the complex of studies conducted using this cell line is quite acceptable for publication, as it answers the question posed. 2. The study of the pathways of carcinogenesis is an extremely topical topic. Numerous works are largely descriptive, while the authors of this article focused on a specific signaling pathway molecule and showed the effects of its inhibition. 3. The complex of methods of inhibitory analysis, Western blotting, immunocytochemistry and microscopy is very well and convincingly applied in the work. Based on the totality of modern methods and the results obtained with their help, the article may be worthy of publication in the desired journal. 4. The material and methods are described in detail and the reader immediately understands the complexity of the study. Of course, it would be nice to conduct a study using several cancer cell lines. However, this would increase the size of an already large article. It would be appropriate to use at least the widespread HEK-293 cell model as a control, which, however, cannot be considered an absolutely “normal” line either. 5. As I already indicated in my review, the conclusions are based on the results obtained and do not raise doubts 6. Links are provided to contemporary literature. The discussion of the results is well written and does not need to be expanded or shortened. 7. Now it is more common to write signalling as signaling. I would recommend the authors to work on the abstract. First of all, open the phrase - All together these observations indicate that the inhibition of a single oncoadaptor protein could exert paradoxical effects. Paradoxical effects should be clearly listed.
Author Response
Answer to Reviewer 2
Comments and query
This is amazing work.
The article uses modern methods.
All results are presented in an excellent way, the voluminous supplement contributes to the understanding of the work done.
In view of the large amount of material received, the authors must present the results of the work in the form of a graphical figure.
I have done additional peer review of the article as requested. Iam sure that this work is worthy of publication.
- The study "ERK signaling pathway is constitutively activated in NT2D1 non-seminoma cells and modulated by c-MET/HGF axis" is devoted to the regulatory role of this signaling cascade in the pathophysiology of cancer cells. The authors used one cell line (NT2D1 embryonal carcinoma cells). However, the complex of studies conducted using this cell line is quite acceptable for publication, as it answers the question posed.
- The study of the pathways of carcinogenesis is an extremely topical topic. Numerous works are largely descriptive, while the authors of this article focused on a specific signaling pathway molecule and showed the effects of its inhibition.
- The complex of methods of inhibitory analysis, Western blotting, immunocytochemistry and microscopy is very well and convincingly applied in the work. Based on the totality of modern methods and the results obtained with their help, the article may be worthy of publication in the desired journal.
ANSWER 1-3: we wish to thank the reviewer for appreciating the work done and for the suggestions. A graphical abstract has been provided to describe the main message of this study.
- The material and methods are described in detail and the reader immediately understands the complexity of the study. Of course, it would be nice to conduct a study using several cancer cell lines. However, this would increase the size of an already large article. It would be appropriate to use at least the widespread HEK-293 cell model as a control, which, however, cannot be considered an absolutely “normal” line either.
ANSWER 4: we wish to thank the reviewer for the suggestion. It was not our intention to compare normal and malignant cells, even because it is not possible to culture normal mitotically active male germ cells for days (just for few hours). Currently, we are exploring in parallel the behaviour and pathways of TCam2 seminoma cells, but the results obtained indicate that the seminoma cells respond to tyrosine kinases activating completely different biological responses, and therefore the results on other TGCT-derived cell lines are not comparable with the result reported in this manuscript.
- As I already indicated in my review, the conclusions are based on the results obtained and do not raise doubts
- Links are provided to contemporary literature. The discussion of the results is well written and does not need to be expanded or shortened.
ANSWER 5-6: we wish to thank the reviewer for appreciating the work done.
- Now it is more common to write signalling as signaling. I would recommend the authors to workon the abstract. First of all, open the phrase - All together these observations indicate that the inhibition of a single oncoadaptor protein could exert paradoxical effects. Paradoxical effects should be clearly listed.
ANSWER 7: we wish to thank the reviewer for the suggestion. The abstract has been changed accordingly.
Reviewer 3 Report
In the present manuscript authors anayzed the modulation of ERK signaling by HGF in non-seminoma cells.
My suggestions:
-improve the background in abstract section;
-specify the meaning of all acronyms used the first time they are mentioned;
- The quality of the images showing WB analysis is low; use low exposure film in figure 1.
- A WB analysis with c-MET should be added
- Perform proliferation analysis at different time points
- What is the possible mechanism by which HGH negatively control cell migration?
- The quality of image in figure 6 should be ameliorated
- Explain better the rationale of paragraph 3.7. It seems not well linked to the main topic of the manuscript.
- Add tests assessing EMT transition
Results should be confirmed by using a HGF antagonist or by silencing the HGF receptor.
Author Response
Answers to Reviewer 3
Comments and Suggestions for Authors
In the present manuscript authors analyzed the modulation of ERK signaling by HGF in non-seminoma cells.
My suggestions:
- improve the background in abstract section;
ANSWER: we wish to thank the reviewer for the suggestion. The abstract background has been expanded.
- specify the meaning of all acronyms used the first time they are mentioned;
ANSWER: the text has been thoroughly checked and the meaning of all the acronyms has been mentioned at the first time they were used.
- The quality of the images showing WB analysis is low; use low exposure film in figure 1.
ANSWER: The western blot analysis has been repeated and the blots of figure 1 have been changed.
- A WB analysis with c-MET should be added
ANSWER: the expression analysis of c-Met receptor on TGCT derived cell lines has been provided in our previous papers on TGCTs (see reference 4, 5, and 8)
- Perform proliferation analysis at different time points
ANSWER: the decision to perform the proliferation assay after 48 h depended on previous articles (see reference 4, by which it is evident that, due to the relatively low proliferation rate of this cells, 48 h is the best time to evaluate HGF- triggered cell proliferation.
- What is the possible mechanism by which HGH negatively control cell migration?
ANSWER: Actually, we found that HGF positively control, cell chemotaxis and collective migration. We understand that the graph of figure 5D can be misleading because the percentage of open residual area decrease when cell migrate. We better describe this result in the revised version of the manuscript and the Y axis of Figure 5 D has been changed with the “percentage of closed area".
- The quality of image in figure 6 should be ameliorated
ANSWER: The images of the figure have been improved, and the Confocal colocalization analysis of F-actin and vinculin has been added.
- Explain better the rationale of paragraph 3.7. It seems not well linked to the main topic of the manuscript.
ANSWER: we better describe the rationale of paragraph 3.7.
- Add tests assessing EMT transition
ANSWER: In the text we stated a partial EMT in HGF and UO126 treated samples. As far as we know the study of the EMT markers together with the study of cell shape are good indicators of EMT. However, to better asses partial EMT we added in Figure 5 a new panel (E) that describe F-actin distribution in cell migrating in wound healing assay after 24h of culture. By these images is clear that in treated samples cells migrate both collectively and as single cells indicating that some of them have done EMT, whereas part of them maintains at least partially their epithelial feature.
Results should be confirmed by using a HGF antagonist or by silencing the HGF receptor.
ANSWER: the results obtained by using c-MET selective inhibitor PF-04217903 has reported in previous papers (see reference number 4, and 8).
Round 2
Reviewer 1 Report
This revised manuscript can be accepted. No further comments.